# Comparisons of the Oral Microbiota from Seven Species of Wild Venomous Snakes in Taiwan Using the High-Throughput Amplicon Sequencing of the Full-Length 16S rRNA Gene

**DOI:** 10.3390/biology12091206

**Published:** 2023-09-04

**Authors:** Wen-Hao Lin, Tein-Shun Tsai

**Affiliations:** 1Institute of Wildlife Conservation, National Pingtung University of Science and Technology, Pingtung 912301, Taiwan; chrislin840518@gmail.com; 2Department of Biological Science and Technology, National Pingtung University of Science and Technology, Pingtung 912301, Taiwan

**Keywords:** wild snakes, bacteriomics, pathogens, next-generation sequencing, conservation medicine, clinical treatment, human health, metagenomics

## Abstract

**Simple Summary:**

There are millions of snake envenoming cases annually worldwide, which is an important but often neglected tropical disease. The morbidity and mortality from snake bites are not only caused by the venom toxins but also the secondary infection from bacteria from the snake itself or an external source. Therefore, if we do not understand the actual pathogens of infection, the expected treatment efficacy is reduced. In this study, we investigated the microbiota carried in the oral cavity of seven wild venomous snakes in Taiwan using full-length 16S rRNA next-generation sequencing. The results showed that non-pathogenic bacteria and pathogenic bacteria were similar in dominance in the oral cavity of snakes, Gram-negative bacteria were more common than Gram-positive bacteria, and the microbiota were significantly different between the Elapidae and Viperidae families of snakes. Our findings will assist in resolving the diversity of oral microbiomes in snakes and can be applied to the veterinary medicine and clinical therapy of venomous snakebites.

**Abstract:**

A venomous snake’s oral cavity may harbor pathogenic microorganisms that cause secondary infection at the wound site after being bitten. We collected oral samples from 37 individuals belonging to seven species of wild venomous snakes in Taiwan, including *Naja atra* (Na), *Bungarus multicinctus* (Bm), *Protobothrops mucrosquamatus* (Pm), *Trimeresurus stejnegeri* (Ts), *Daboia siamensis* (Ds), *Deinagkistrodon acutus* (Da), and alpine *Trimeresurus gracilis* (Tg). Bacterial species were identified using full-length 16S rRNA amplicon sequencing analysis, and this is the first study using this technique to investigate the oral microbiota of multiple Taiwanese snake species. Up to 1064 bacterial species were identified from the snake’s oral cavities, with 24 pathogenic and 24 non-pathogenic species among the most abundant ones. The most abundant oral bacterial species detected in our study were different from those found in previous studies, which varied by snake species, collection sites, sampling tissues, culture dependence, and analysis methods. Multivariate analysis revealed that the oral bacterial species compositions in Na, Bm, and Pm each were significantly different from the other species, whereas those among Ts, Ds, Da, and Tg showed fewer differences. Herein, we reveal the microbial diversity in multiple species of wild snakes and provide potential therapeutic implications regarding empiric antibiotic selection for wildlife medicine and snakebite management.

## 1. Introduction

There are approximately 2.7 million cases of snake envenoming annually worldwide, resulting in 81,000–138,000 deaths [1]. The World Health Organization (WHO) placed it in the category A of the neglected tropical diseases in 2017 [2,3,4]. The morbidity and mortality from snakebites are caused by not only the venom toxins but also the secondary infection from pathogenic bacteria [5,6,7]. There may be abundant microorganisms within a snake’s oral cavity, including harmless environmental bacterial species and opportunistic, human-pathogenic species [8,9]. After a snakebite, the venom causes bruising around the bite site, and abscesses, pustules, or necrosis can develop, which creates a suitable environment for the bacteria from the snake’s oral cavity and saliva [4,7,10,11,12]. Many bacterial species could cause secondary infection, including Gram-positive, Gram-negative, and anaerobic bacteria. Anaerobic bacteria can cause gas gangrene after infecting the bite wound; Gram-positive bacteria can cause infection symptoms, such as an abscess, cellulitis, sepsis, meningitis, and urinary tract infections; and Gram-negative bacteria can cause several infection symptoms, such as gastroenteritis, sepsis, respiratory infection, meningitis, diarrhea, fever, and soft tissue infection [7,8,12,13,14,15,16,17,18]. Although anti-serum therapy can reduce venom toxicity to human tissue and the organ system, it cannot prevent a secondary infection caused by bacteria. The treatment for bacterial infections is typically empiric antibiotic administration, but the microbiota of the infected wound and oropharynx of the culprit snakes must be properly established. If the actual pathogens of infection are unknown, the expected treatment efficacy would decrease, and antibiotic resistance would likely be induced [4,9,19].

In addition to potentially infecting humans by snakebites, the oral bacteria of snakes also affect snakes themselves [16,20,21]. Snakes may experience oral wounds or infection when feeding prey or contacting foreign substances in the environment [22,23,24]. The bacteria may invade through the respiratory and gastrointestinal tracts, causing systemic infection and even death [25,26,27]. In reptiles, Gram-negative bacteria, such as *Pseudomonas*, *Aeromonas*, *Proteus*, and *Escherichia*, may be common taxa that cause infection [24,27]. *Pseudomonas aeruginosa*, *Providencia rettgeri*, and *Stenotrophomonas maltophilia* are often the dominant bacteria in the oral cavity of snakes suffering from stomatitis. In addition, *Klebsiella*, *Acinetobacter*, *Citobacter*, *Chlamydia*, *Morganella*, *Staphylococcus*, *Bacteroides*, *Fusobacterium*, *Clostridium*, *Mycobacterium*, and *Peptostreptococcus* are also common pathogenic bacteria in the oral cavity of snakes [24,25]. Previous studies showed that the oral cavity microbiomes may vary by the species, season, habitat, health situation, and predation strategy of snakes [7,9,12,13,16,17,20,28,29]. Even if the phylogenetic relationship among snake species is close or they are of the same species, they may have different oral cavity bacterial species if their habitat and activity range are not similar [9,28]. In terms of the development of wildlife medicine, only by better understanding the oral bacteria can the subsequent therapy and empiric antibiotic selection be precisely and efficiently made on snakes.

Owing to the lack of information on the complete microbiota in the oral cavity of Taiwanese snake species (except [30]) and their relationship with environmental factors, this study conducted high-throughput sequencing (next-generation sequencing; NGS) of the full-length 16S rRNA gene to identify the full compositions of microbiota within the oral cavity of seven Taiwanese snake species belonging to the Viperidae and Elapidae families. The nine hypervariable regions (V1–V9) of bacterial 16S ribosomal RNA genes can be targeted to identify bacterial taxa in 16S amplicon NGS studies [31,32], in contrast to the short-read NGS used in some previous studies, which used short fragments, such as V3–V4 regions, as target references [33]. Without the sequence information from other hypervariable regions, information at the species level usually cannot be fully acquired by short-read NGS, whereas long-read sequencing can fix the issues revealed with short-reads, such as genome-wide repeats and structural variant detection [34]. In this study, we conducted long-read sequencing by virtual PCR with primers covering the V1–V9 region of the 16S rRNA gene as references. We aimed to investigate not only the diversity of snake oral cavity microbiota within and among species, but also their correlations with altitude. We also discussed the results of previous studies compared to the present study, regarding different snake species, collection sites (in clinic, out-of-hospital, in animal house, or in wild), sampling tissues (oral cavity, gut, or bite wound), culture dependence, and analysis methods (CBtest, mass spectrometry, or 16S rRNA sequencing).

## 2. Materials and Methods

### 2.1. Sample Collection

A total of 37 snakes from seven venomous species were sampled in this study, including *Naja atra* (Na; *n* = 6), *Bungarus multicinctus* (Bm; *n* = 5), *Protobothrops mucrosquamatus* (Pm; *n* = 6), *Trimeresurus stejnegeri* (Ts; *n* = 5), *Daboia siamensis* (Ds; *n* = 5), *Deinagkistrodon acutus* (Da; *n* = 5) and alpine *Trimeresurus gracilis* [35] (Tg; *n* = 5) (Figure 1a–g), which were captured from the wild in Kaohsiung, Pingtung, Taitung, Yilan, and Nantou in Taiwan (Figure 2). To avoid the original microbiota within the oral cavity being affected by traffic and transporting procedures, we sampled the snakes in as soon as possible (2.4 ± 2.3 [mean ± SD] days) after collection from the wild. To prevent cross-contamination, we opened the mouth of the snake using a sterilized, hollowed out wooden stir stick and used sterilized cotton-tipped swab sticks to collect oral samples (Figure 1h) [36]. Samples were taken by rotating the cotton tip of the swab on the floor of the oral cavity between the larynx and mandibular teeth from each snake after securing the head [12]. Subsequently, sample swabs were placed separately into sterilized tubes and stored immediately at −80 °C for later analysis. Snakes were released back into the wild after sampling.

### 2.2. DNA Extraction, PCR Amplification, and Purification

DNA was extracted by organic extraction (chloroform: isoamyl alcohol = 96:4) [37], and DNA concentration was determined using a Qubit 4.0 Fluorometer (Thermo Scientific, Waltham, MA, USA) and adjusted to 1 ng/μL for the following process. The full-length 16S genes (V1–V9 regions) were amplified by barcoded 16S gene-specific primers. According to the Amplification of Full-Length 16S Gene with Barcoded Primers for Multiplexed SMRTbell Library Preparation and Sequencing Procedure (PacBio, Menlo Park, CA, USA), each primer was designed to contain a 5′ buffer sequence (GCATC) with a 5′ phosphate modification, a 16-base barcode, and the degenerate 16S gene-specific forward or reverse primer sequences (Forward: 5′Phos/GCATC-16-base barcode-AGRGTTYGATYMTGGCTCAG-3′, Reverse: 5′Phos/GCATC-16-base barcode-RGYTACCTTGTTACGACTT-3′). Degenerate base identities are as follows: R = A, G; Y = C, T; M = A, C. Briefly, 2 ng of gDNA was used for the PCR reaction performed with KAPA HiFi HotStart ReadyMix (Roche, Basel, Switzerland) under the following PCR conditions: 95 °C for 3 min; 20–30 cycles (depending on the sample) at 95 °C for 30 s, 57 °C for 30 s, 72 °C for 60 s; 72 °C for 5 min, and hold at 4 °C. The PCR products were monitored on 1% agarose gel. Samples with bright main strip of approximately 1500 bp were chosen and purified using the AMPure PB beads for the following library preparation.

### 2.3. SMRTbell Library Construction and Sequencing

The SMRTbell library was prepared according to PacBio. Briefly, equal molars of each barcoded PCR product were pooled, and 500–1000 ng of the pooled amplicon sample was used for DNA damage repair followed by end repair/A-tailing and ligation steps to introduce the universal hairpin adapters onto double-stranded DNA fragments. After purification with AMPure PB beads to remove the adapter dimer, the SMRTbell library was incubated with Sequel II primer 3.1 and Sequel II Binding Kit 3.1 (PacBio, Menlo Park, CA, USA) for the primer annealing and polymerase binding. Finally, sequencing was performed in the circular consensus sequence (CCS) mode on a PacBio Sequel IIe instrument (PacBio, Menlo Park, CA, USA) to generate the HiFi reads with predicted accuracy (Phred Scale) = 30.

### 2.4. Data Analysis

The CCS reads were determined with a minimum predicted accuracy of 0.9 and the minimum number of passes set to 3 in the official workflow of PacBio through the SMRT Link software (version 11.1). Only CCS reads with a quality score greater than Q30, referred to as Q30 HiFi reads, were used in the next stage. After demultiplexing, the HiFi reads were further processed with DADA2 (version 1.20) to obtain amplicons with single-nucleotide resolution [38,39]. The trimming and filtering were performed with a maximum of two expected errors per read (maxEE = 2). DADA2 algorithm resolves exact amplicon sequence variants (ASVs) with single-nucleotide resolution from the full-length 16S rRNA gene with a near-zero error rate. For each representative sequence, the feature-classifier [40] and classify-consensus-vsearch [41] algorithm in QIIME2 (version 2022.11) [42] was employed to annotate the taxonomy classification based on the information retrieved from the NCBI 16S ribosomal RNA database. To analyze the sequence similarities among different ASVs, multiple sequence alignment was conducted using the QIIME2 alignment, MAFFT [43], against the NCBI 16S ribosomal RNA database [44,45,46]. In addition, the estimation of the abundance was deduced from the assigned read counts of the sequences (=non-chimeric read counts − unassigned read counts); the relative abundance was calculated from the assigned read counts for each snake or bacteria species divided by the total assigned read counts.

Several alpha-diversity indices were used to measure alpha diversity for each sample, including observed-species, Menhinick’s richness, Margalef’s richness, Shannon, Simpson, Pielou’s evenness, PD whole tree, and Good’s coverage [47]. All indices were calculated with QIIME. Kruskal–Wallis tests were conducted to test the differences on the alpha diversity of oral bacteria among snake species. The “ggplot2” package in R (version 3.6.0) was used to generate the plots, including rarefaction curves. Beta diversity analysis was also applied to analyze the difference in species complexity between samples. An UpSet plot was generated using the “UpSetR” package in R to present the number of common and uniquely identified bacterial species in the samples among snake species. Heatmap representation was generated using the “ggplot2” package in R to show the abundance distribution of the top 35 bacterial genera or species identified among the individual oral samples of different snake species. Dissimilarity matrices (Bray-Curtis and weighted Unifrac) [48,49] were generated using the “micro-eco” package in R. To visualize patterns in multidimensional data, the dimensionality reduction method of constrained principal co-ordinates analysis (CPCoA) [50,51] was conducted using the “micro-eco” and “vegan” packages in R. The *p*-value of CPCoA test was calculated using the “anova.cca” permutation package in R. In addition, we used UPGMA to construct a tree that reflects the similarities between samples presented in the UniFrac distance matrix. “ggplot2” and “plotly” were used to generate the plots. For statistical analysis, the significance of all species among groups at various taxonomic level was detected using differential abundance analysis with a zero-inflated Gaussian (ZIG) log-normal model as implemented in the “fitFeatureModel” function of the Bioconductor “metagenomeSeq” package in R [52]. Moreover, Welch’s *t*-test was performed using the “stat” package in R to compare metagenomic profiles of the oral bacterial communities among snake species.

## 3. Results

### 3.1. Sequence Analysis

Total non-chimeric reads of 71,848 (unassigned reads 670), 54,328 (unassigned reads 0), 68,820 (unassigned reads 35), 80,195 (unassigned reads 40), 61,205 (unassigned reads 105), 55,877 (unassigned reads 145), and 56,836 (unassigned reads 109) were obtained for Na, Bm, Pm, Ts, Ds, Da, and Tg samples, respectively (Table 1). After DADA2 denoising, full-length 16S rRNA sequences for each sample were generated. From a total of 3241 ASVs, 49 were removed because the species was unassigned, and 3192 were considered for further analysis. The rarefaction plot (Appendix A) reveals that the number of observed bacterial species plateaus when the sequencing depths reach 10,000 in most samples.

### 3.2. Taxonomic Profiling of Metagenomic Sequences

We discovered 163, 207, 204, 192, 478, 253, and 471 bacterial species in Na, Bm, Pm, Ts, Ds, Da, and Tg, respectively (Table 2). At the phylum level, Proteobacteria, Bacteroidetes, and Firmicutes were commonly distributed among the samples (Figure 3a), while the Tenericutes, Actinobacteria, Fusobacteria, Cyanobacteria, Verrucomicrobia, and Planctomycetes did not present in some parts of samples (Figure 4a). Proteobacteria (Na: 65.3%, Bm: 62.7%, Pm: 37.6%, Ts: 40.1%, Ds: 59.5%, Da: 70.5%, Tg: 58.9%) and Bacteroidetes (Na: 15.3%, Bm: 18.0%, Pm: 18.9%, Ts: 58.6%, Ds: 15.9%, Da: 19.3%, Tg: 12.3%) were identified as the dominant phyla (Figure 3a). At the species level, *Puia dinghuensis*, *Cupriavidus numazuensis*, *Mycoplasma fastidiosum*, *Erysipelatoclostridium innocuum*, *Bacteroides fragilis*, *Morganella morganii*, *Haemophilus felis*, *Citrobacter freundii*, *Stenotrophomonas maltophilia*, and *Bordetella trematum* are the 10 most abundant species as a whole, which accounted for 51.71 ± 21.03% of the total (mean ± SD; Figure 3b), while the 20 most abundant species accounted for 67.28 ± 18.31% of the total. None of the 10 most abundant species presented in all individual samples (Figure 4b); the corresponding results for each snake species are shown in Table 3.

Taxonomy annotations and relative abundances of the bacteria are summarized in Appendix A. A total of 44 unique ASVs with 1104 reads are reported as unassigned. Of the 448,005 total reads, we identified 1064 unique species among the 37 samples. The number of common and unique bacterial species among seven snake species is shown in Figure 5. Seven bacterial species (eight ASVs) were shared by all snake species, including *Methylibium petroleiphilum*, *Delftia lacustris*, *Caulobacter segnis*, *Pedobacter nutrimenti*, *Cupriavidus numazuensis*, *Sphingomonas paucimobilis*, and *Brevundimonas diminuta*. In addition, 22 species (207 ASVs), 32 species (230 ASVs), 39 species (277 ASVs), 39 species (197 ASVs), 193 species (798 ASVs), 108 species (337 ASVs), and 189 species (591 ASVs) were unique to Na, Bm, Pm, Ts, Ds, Da, and Tg, respectively. The summary of the alpha diversity results is shown in Appendix A. The alpha diversity of oral bacteria in Ds was the highest (e.g., Kruskal–Wallis test: *p* = 0.037 for Menhinick’s richness index).

Furthermore, we also presented the abundance distribution of the top 35 bacterial genera and species among individual oral samples of the seven snake species in heatmaps (Figure 6). The abundance patterns of dominant bacterial genera or species varied in the samples among snake species and were more consistent within the same snake species. At the genus level, *Bordetella* was the most abundant, followed by *Pseudomonas* and *Mycoplasma* in Na; *Haemophilus*, *Mycoplasma*, and *Stenotrophomonas* were the most abundant in Bm; *Erysipelatoclostridium*, *Klebsiella*, and *Puia* in Pm; *Puia*, *Citrobacter*, and *Bacteroides* in Ts; *Cupriavidus*, *Romboutsia*, and *Salmonella* in Ds; *Morganella*, *Puia*, and *Providencia* in Da; *Cupriavidus*, *Mycoplasma*, and *Pseudomonas* in Tg (Figure 6a).

Among the top 10 abundant bacterial species in the oral cavity of seven snake species, totally, 24 species have been reported as pathogens or opportunistic pathogens to humans, while 24 species had no pathogenicity toward humans, although they might have been isolated from an infection in a human, and most of them are non-spore-forming bacteria (Table 4). For the top 10 most abundant bacterial species in all snake species, five were pathogenic, four were non-pathogenic, and one remains to be elucidated; one was Gram-positive, and nine were Gram-negative; and six were aerobic (two were facultative anaerobic), and two were anaerobic.

### 3.3. Comparison of Bacterial Community Structure

The CPCoA analysis revealed that Ts, Ds, Da, and Tg clustered closely by sharing identical ASVs at the species level, whereas the bacterial species identified in Na, Bm, and Pm were uniquely distributed and significantly deviated from Ts, Ds, Da, and Tg (*p* = 0.001; Figure 7).

Comparison of a metagenomic profile using Welch’s *t*-test revealed that there are significant differences in abundances of at least one bacterial taxon between Na vs. Bm, Na vs. Pm, Na vs. Ts, Na vs. Ds, Na vs. Da, Na and Tg, Bm vs. Pm, Bm vs. Ts, Bm vs. Ds, Bm vs. Da, Bm vs. Tg, Pm vs. Da, Ts vs. Ds, and Ds vs. Da samples (Figure 8). Among the taxa with a mean proportion >1%, *Proteiniphilum acetatigenes* was more abundant in Na than in other snake species (all *p*s < 0.05; Figure 8a–f). *Haemophilus felis* was more abundant in Bm than in other snake species (all *p*s = 0.0275; Figure 8a,g–k). *Mycoplasma fastidiosum* was more abundant in Na than in Pm, Ts, and Ds (all *p*s < 0.03; Figure 8b–d). *Stenotrophomonas maltophilia* in Da was less abundant than in Na (*p* = 0.0345; Figure 8e) or Pm (*p* = 0.0424; Figure 8l).

## 4. Discussion

In Taiwan, previous studies on the bacterial species in a snake’s oral cavity were much fewer than those in clinical wound samples. The methods to identify the bacterial species usually included bacterial culture and biochemical testing (CBtest), Sanger sequencing, and Vitek2 system for culturing bacteria [15,33]. However, not all bacteria are easy or able to be cultured, and the bacterial compositions after being cultured may differ from the original ones. In addition, for the 16S rRNA NGS used to investigate the microbiota in a snake’s oral cavity, the precision and accuracy of bacterial identification would be reduced if full-length variable regions are not considered as a sequencing target [34]. Therefore, the bacterial identification results in this study are more robust as we investigated the oral bacterial community composition using full-length 16S rRNA amplicon sequencing analysis.

Our results showed that Proteobacteria and Bacteroidetes were the most dominant phyla among the snakes analyzed. In a previous study, the amplicon 16S rRNA gene V3–V4 hypervariable region showed that Proteobacteria and Actinobacteria are the dominant phyla in the oral bacterial community of *Naja naja*, *Ophiophagus hannah*, and *Python molurus* [36]. Another study using the V4 region of the 16S rRNA gene [53] showed the dominant phyla are Proteobacteria, Firmicutes, and Actinobacteria in saliva samples of Komodo dragons. In the present study, Actinobacteria were not listed in the top three phyla in our results. In addition, previous bacteriomic studies on a snake’s gastrointestinal tract using 16S rRNA gene high-throughput sequencing demonstrated that Proteobacteria and Bacteroidetes are the dominant bacterial phyla in *Naja atra*, *Ptyas mucosa*, *Elaphe carinata,* and *Deinagkistrodon acutus* [54,55], as well as *Rhabdophis subminiatus* [56]. Another gut bacteriomic study on sea snakes (*Hydrophis curtus* and *Hydrophis cyanocinctus*) using the amplicon V3–V4 regions of the 16S rRNA genes also showed that Proteobacteria and Bacteroidetes were the most abundant phyla [57]. Thus, the dominant bacterial phyla in the present study are similar to those in the studies on bacterial communities in the gastrointestinal tract of reptiles.

The most previous studies in Taiwan collected samples of infected tissue or necrosis sites from the wounds of snake-bite patients and identified the bacterial composition to the genus or species level using the CBtest [33,58]. More research was conducted on common venomous species, like Na, Pm, and Ts, and indicated that *Enterococcus faecalis* and *Morganella morganii* are the most dominant bacterial species [59,60,61,62], followed by *Aeromonas hydrophila*, *Serratia marcescens*, *Proteus vulgaris*, *Bacteroides fragilis*, and *Pseudomonas aeruginosa* [15,58]. Another study on Pm and Ts showed the dominant bacterial species were *Enterococcus faecalis* and *Morganella morganii*, followed by *Staphylococcus* spp., *Corynebacterium* spp., and *Enterobacter* spp. [63]. In addition, a recent study, which sampled the snake’s oral cavity and not bite tissue, used a cultivation and matrix-assisted laser desorption/ionization time-of-flight mass spectrometry analysis, which showed that the most common bacterial species harbored in Na, Bm, Ts, and Pm of southern Taiwan were *Enterococcus faecalis*, *Pseudomonas aeruginosa*, *Clostridium* spp., *Bacteroides fragilis*, *Proteus vulgaris*, and *Citrobacter freundii* [64]. In contrast to the above studies, our results (Table 3) identified different dominant bacteria, which were *Bordetella trematum*, *Mycoplasma fastidiosum*, and *Phocoenobacter uteri* in Na (*Morganella morganii* ranked 9th); *Haemophilus felis*, *Stenotrophomonas maltophilia*, and *Puia dinghuensis* in Bm; *Clostridium innocuum*, *Cupriavidus numazuensis,* and *Puia dinghuensis* in Pm; and *Puia dinghuensis*, *Bacteroides fragilis,* and *Citrobacter freundii* in Ts (*Morganella morganii* ranked sixth). Rather, *Morganella morganii* was the most dominant in Da. *Enterococcus faecalis* was not in the top 10 species for all snake species but ranked 24th, 50th, 130th, and 200th in Na, Ds, Da, and Tg, respectively.

Furthermore, several reports have presented potential advantages of molecular diagnostics over microbial culture, namely, a shorter turnaround time, detection of bacteria difficult to culture, or detection after prior administration of antibiotics that inhibit bacterial growth [33,65]. The advantage of the NGS-method 16S metagenomics assay is not only the identification of species but also the quantification of the relative abundance of all bacteria in polymicrobial infections [33]. Previous results from a culture-based method in Na only detected 2% of the bacterial species identified using the 16S rRNA NGS method. The most abundant species in Na observed using the 16S rRNA method were different from those determined using culture-based methods in [33], which revealed that the most common bacterial species in the oropharynx of Na were *Pseudomonas azotoformans*, *P. lundensis*, *Delftia tsuruhatensis*, and *Methylobacterium goesingense*, where only *P. azotoformans* was also dominant in our study. The variations of bacterial species identified among the different studies may result from the differences in snake species, collection sites (in clinic, out-of-hospital, in animal house, or in wild), sampling tissues (oral cavity, gut, or bite wound), culture dependence, and analysis methods (CBtest, mass spectrometry, or 16S rRNA sequencing).

In our study, *Puia dinghuensis* was the most abundant bacteria species (ranked in the top 10) among all snake species except Ds (Table 3), but it is not a pathogenic or opportunistic-pathogenic bacterium to humans and is typically isolated from the monsoon evergreen broad-leaved forest soil [66]. Most previous studies focused on pathogenic bacterial species carried in a snake’s oral cavity. Among the seven bacterial species shared by all snake species in our study, only *Delftia lacustris* and *Sphingomonas paucimobilis* have been reported as pathogenic, and both are Gram-negative bacteria [67,68,69]. The five remaining bacterial species are all non-pathogenic, including *Methylibium petroleiphilum*, a Gram-negative bacterium, which prefers aerobic, warm, and close to neutral pH conditions [70,71]; *Caulobacter segnis*, a Gram-negative bacterium, which is typically isolated from soil [72,73]; *Pedobacter nutrimenti*, a Gram-negative bacterium and non-spore-forming bacterial strain that is typically isolated from soils, other environmental sources, or from food [74]; *Cupriavidus numazuensis*, a Gram-negative bacterium typically isolated from soil [75]; *Brevundimonas diminuta*, a Gram-negative bacterium with a worldwide distribution and is typically isolated from several sites, including water, soil, and plants [76,77].

Generally, bacterial isolates from the oral cavity of snakes are not all pathogenic: They may cause wound infections or abscesses in humans but so do normal environmental contaminants or soil pathogens [33]. Most of pathogenic bacterial species are Gram-negative [7]. Five bacterial species in the top 10 abundant bacterial species found in our study have been reported as pathogenic, including *Erysipelatoclostridium innocuum*, a Gram-positive bacterium common in intestinal flora [78]; *Bacteroides fragilis*, a Gram-negative bacterium commonly found as part of the normal microbiota of the human colon; *Morganella morganii*, a Gram-negative bacterium found in the feces and intestines of humans, dogs, and other mammals [79,80]; *Citrobacter freundii*, a Gram-negative bacterium found in water, soil, food, and the intestines of humans and other animals [81,82]; *Stenotrophomonas maltophilia*, a ubiquitous Gram-negative bacterium frequently isolated from soil, water, animals, plant matter, and hospital equipment [83,84]. In addition, the pathogenic potential and infection virulence of *Bordetella trematum*, commonly isolated from human wounds and ear infections [85], remains to be elucidated [86].

Our results showed that Elapidae snakes, including Na and Bm, had noticeably different oral cavity microbiota from each other and from the species of Viperidae, which may have similar microbiota among each other. It is known that a Na bite often causes severe tissue necrosis while a Bm bite rarely causes a significant local tissue problem [87,88]. Compared to Ts, victims of Pm have a higher rate (37% vs. 6%) of local necrosis or cellulitis [89]. Victims of Ds, Da, and Tg are rare in Taiwan [90,91], and a Ds bite seems to have less complication of local tissue necrosis [91,92]. Tissue necrosis may be initiated by the action of snake toxins [93,94]. How it is related to the differences of oral bacterial strains between Na and Bm or among Ds, Pm, and other pitvipers needs further studies to verify the ability of the microbiota identified in this study that cause infection and necrosis. The microorganisms harbored by reptiles and their abundances could be affected by their environment [53]. The environment in a snake’s oral cavity is likely affected by the snake’s habitat, home range, and predation strategy [7,16]. We also investigated the oral bacteriome of Tg living at a high altitude in Taiwan to test whether different microbiota could be detected by altitudinal factors; however, the oral bacterial community in this species was not significantly different from the low-altitude species on the existence of confounding factors, such as controversial phylogeny [95]. Further studies on the effects of habitat variations on the metagenomics of microorganisms in snakes are required.

There are several limitations in our research. First, there was usually a delay of several days between snake collection and oral sampling, although we sampled the snakes as quickly as possible after wild collection. Second, there were only five or six snake samples per species, which may not properly represent the wild populations. Third, most of our samples were collected from snakes in southern Taiwan, except for *Trimeresurus gracilis*, which was collected from a high-altitude area around the island; therefore, the collection sites can be expanded in future studies. In addition, although understanding the pathogens from the oral cavities of venomous snakes may assist in empiric antibiotic selection for veterinary medicine and human medicine, physicians still need to consider secondary skin and nosocomial infections after snake envenomation [64]. Our research only focused on venomous snake samples. Most studies focused on venomous snakes because most clinical records are from venomous snake bites. However, pathogens harbored by non-venomous snakes could still be threatening if their teeth are left in the wound or the bite creates a massive open wound. Therefore, assessing the bacterial species carried by non-venomous snakes would fulfill the diversity database of microbiota within a snake’s oral cavity and should be considered in future studies.

## 5. Conclusions

This study is the first to use full-length 16S rRNA next-generation sequencing (NGS) to investigate the complete oral microbiota carried by seven wild snakes in Taiwan. The results showed similar dominance between non-pathogenic bacteria and pathogenic bacteria, and Gram-negative bacteria were more common than Gram-positive ones. The oral microbiota were significantly different between the snakes belonging to Viperidae and Elapidae and between *Naja atra* and *Bungarus multicinctus* within Elapidae, while there was considerable overlap of bacterial species among the species within Viperidae. Our study identified different dominant bacteria compared to previous studies, possibly owing to differences in snake species, collection sites, sampling tissues, culture dependence, and analysis methods. Long-read microbiome analyses of bite wound samples from patients need to be carried out in future studies. This study provides potential therapeutic implications for wildlife medicine and snakebite management.

## Figures and Tables

**Figure 1 biology-12-01206-f001:**
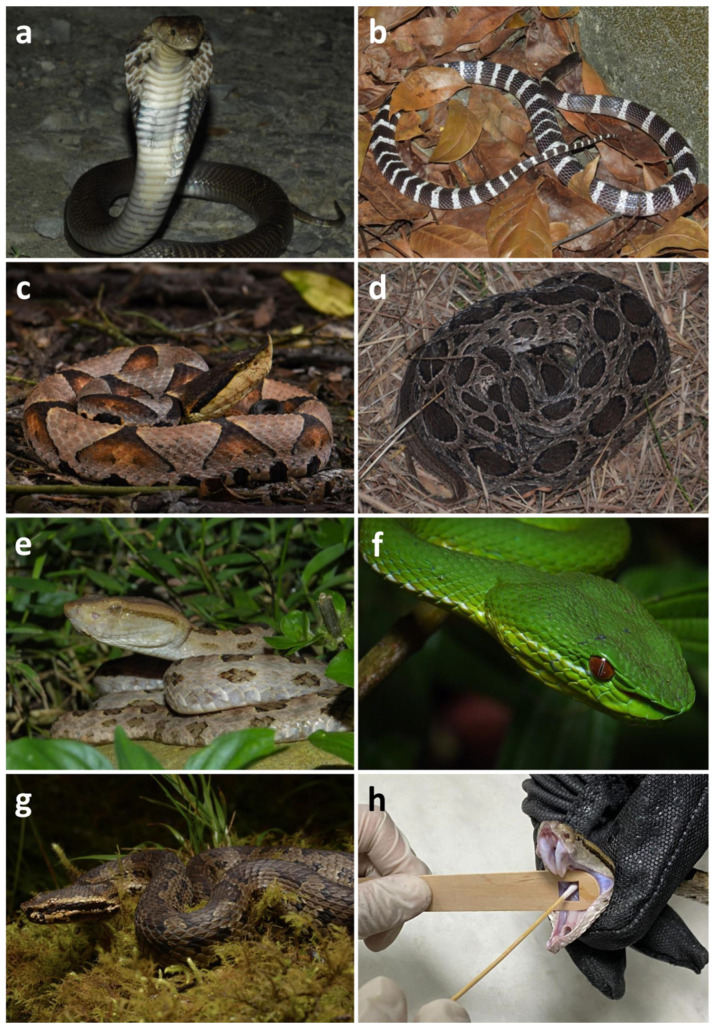
Venomous snakes used for oral sample collection: (**a**) *Naja atra*; (**b**) *Bungarus multicinctus*; (**c**) *Deinagkistrodon acutus*; (**d**) *Daboia siamensis*; (**e**) *Protobothrops mucrosquamatus*; (**f**) *Trimeresurus stejnegeri*; (**g**) *Trimeresurus gracilis*; (**h**) oral sample collection from *P. mucrosquamatus* using sterilized cotton-tipped swab and sterilized, hollowed out wooden stir stick. Photos courtesy of Jun-Wei Zhang (**a**,**b**,**e**), Jui-Hsiang Fan (**c**,**f**), and Tsz-Chun Tse (**d**,**g**).

**Figure 2 biology-12-01206-f002:**
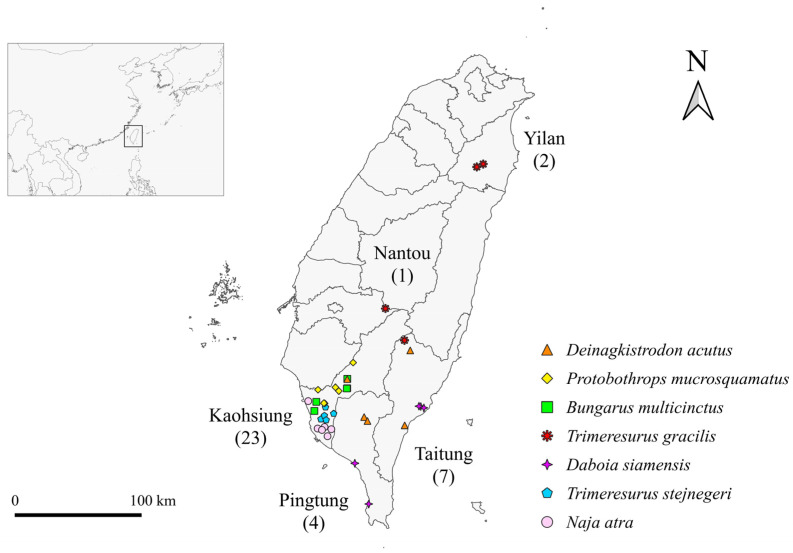
The collection sites and sample sizes of two elapid and five viperid species in five administrative districts of Taiwan.

**Figure 3 biology-12-01206-f003:**
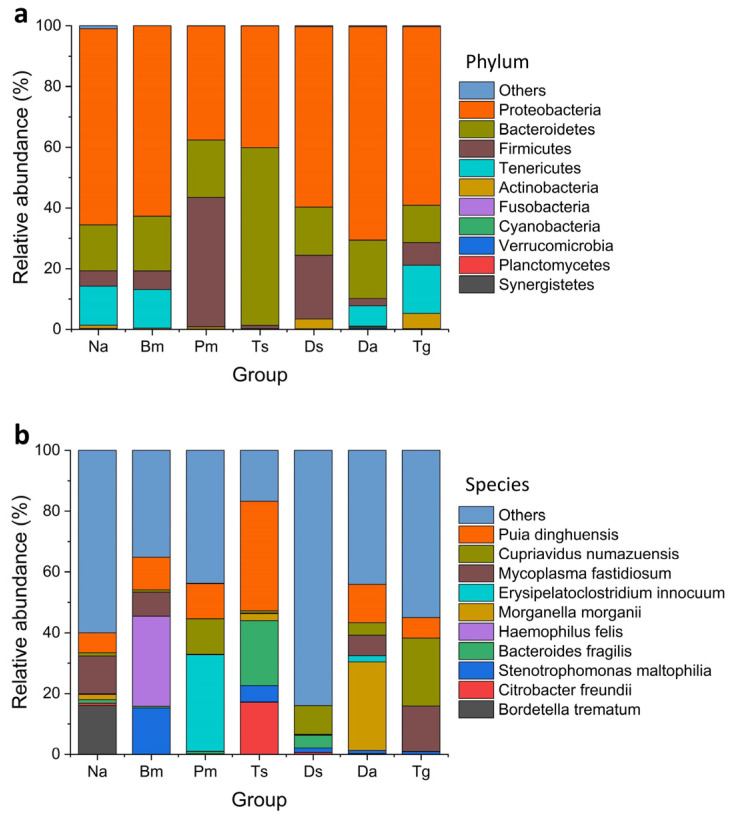
Relative abundances of the dominant bacterial phyla (**a**) and species (**b**) in the oral samples of seven Taiwanese snake species. Those with sequence reads that did not associate with any known reference taxon or the relative abundance is lower than that of the top 10 species were classified as Others. Na: *Naja atra*, Bm: *Bungarus multicinctus*, Pm: *Protobothrops mucrosquamatus*, Ts: *Trimeresurus stejnegeri*, Ds: *Daboia siamensis*, Da: *Deinagkistrodon acutus*, and Tg: *Trimeresurus gracilis*.

**Figure 4 biology-12-01206-f004:**
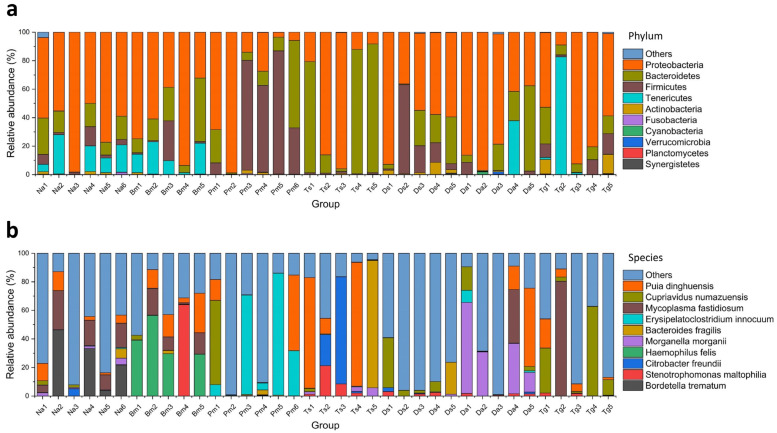
Relative abundances of the dominant bacterial phyla (**a**) and species (**b**) in the individual oral samples of seven Taiwanese snake species. Na: *Naja atra*, Bm: *Bungarus multicinctus*, Pm: *Protobothrops mucrosquamatus*, Ts: *Trimeresurus stejnegeri*, Ds: *Daboia siamensis*, Da: *Deinagkistrodon acutus*, and Tg: *Trimeresurus gracilis*.

**Figure 5 biology-12-01206-f005:**
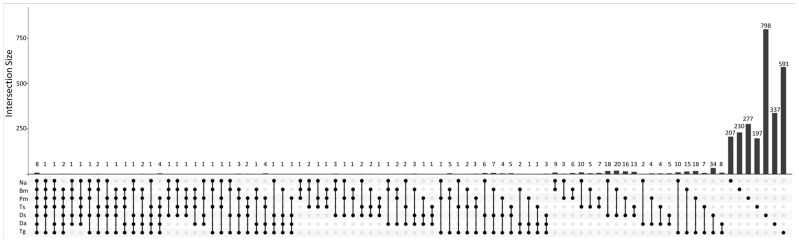
The UpSet plot showing the number of common and uniquely identified bacterial species among the oral samples of seven Taiwanese snake species. For example, for all seven snake species, only seven bacterial species (eight ASVs) were common in all 1064 bacterial species found. Na: *Naja atra*, Bm: *Bungarus multicinctus*, Pm: *Protobothrops mucrosquamatus*, Ts: *Trimeresurus stejnegeri*, Ds: *Daboia siamensis*, Da: *Deinagkistrodon acutus*, and Tg: *Trimeresurus gracilis*.

**Figure 6 biology-12-01206-f006:**
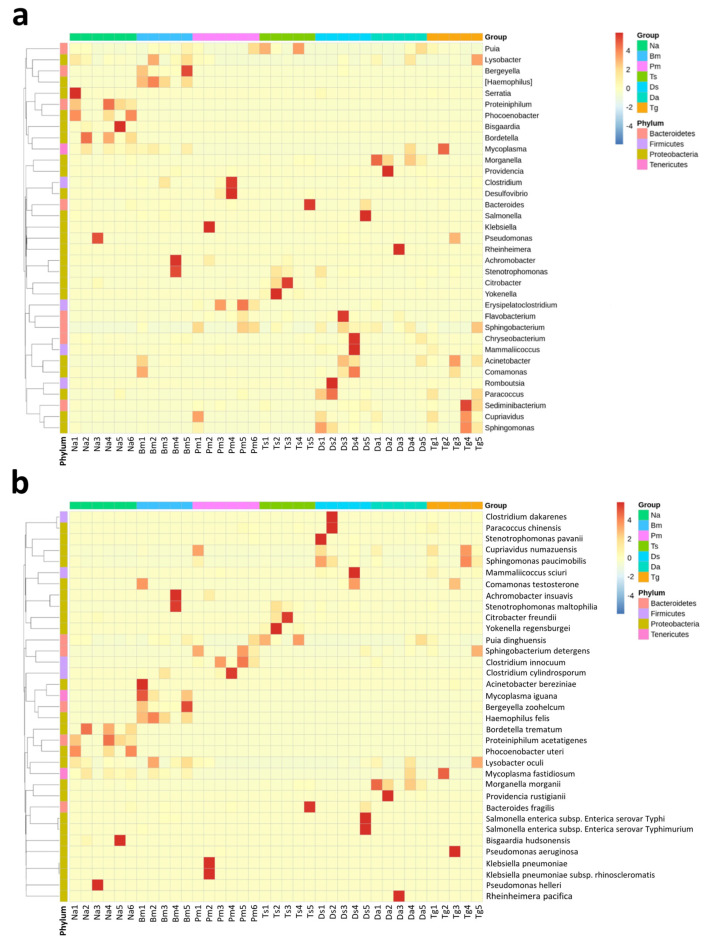
Heatmap showing of the abundance distribution of the top 35 bacterial genera and species identified among the individual oral samples of seven Taiwanese snake species: (**a**) The distribution of the top 35 abundant bacterial genera and (**b**) the top 35 abundant bacterial species among individual snake samples. Na: *Naja atra*, Bm: *Bungarus multicinctus*, Pm: *Protobothrops mucrosquamatus*, Ts: *Trimeresurus stejnegeri*, Ds: *Daboia siamensis*, Da: *Deinagkistrodon acutus*, and Tg: *Trimeresurus gracilis*.

**Figure 7 biology-12-01206-f007:**
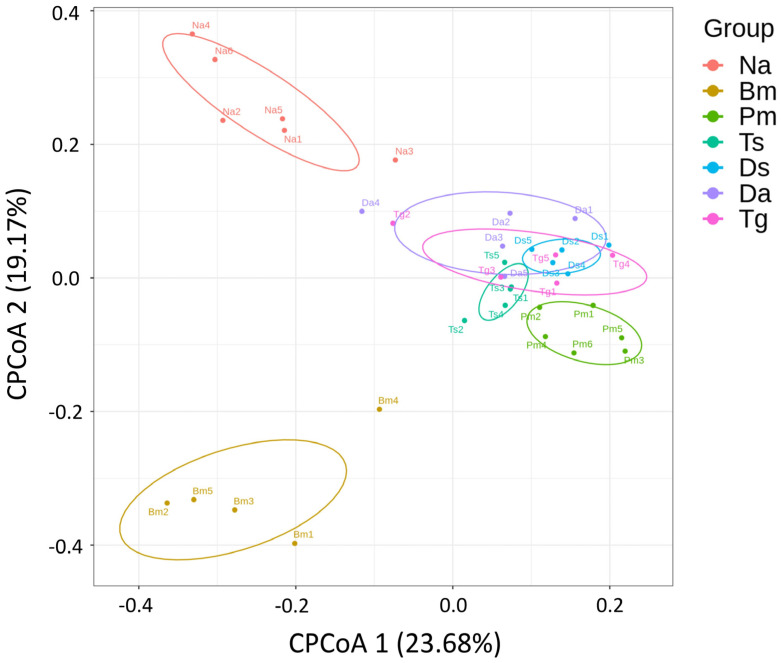
The coordinate plot for the constrained principal coordinates analysis (CPCoA) of the oral bacterial communities among the samples of seven Taiwanese snake species. The percentages in the brackets represent the contribution to sample variation. The significance of the CPCoA calculated from the “anova.cca” processed permutation in R was *p* = 0.001. Na: *Naja atra*, Bm: *Bungarus multicinctus*, Pm: *Protobothrops mucrosquamatus*, Ts: *Trimeresurus stejnegeri*, Ds: *Daboia siamensis*, Da: *Deinagkistrodon acutus*, and Tg: *Trimeresurus gracilis*.

**Figure 8 biology-12-01206-f008:**
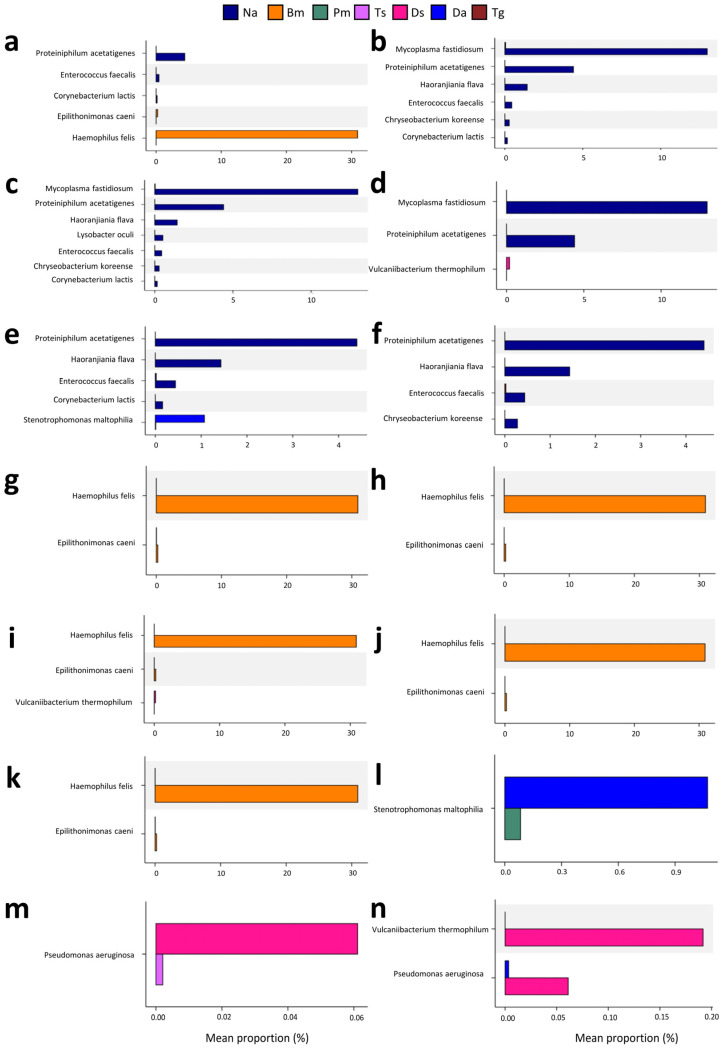
Metagenomic profile comparisons of the oral bacterial communities among the seven Taiwanese snake species, determined using Welch’s *t*-test. Only the taxa abundance comparisons with a significant difference (*p* < 0.05) are shown in (**a**) Na vs. Bm, (**b**) Na vs. Pm, (**c**) Na vs. Ts, (**d**) Na vs. Ds, (**e**) Na vs. Da, (**f**) Na and Tg, (**g**) Bm vs. Pm, (**h**) Bm vs. Ts, (**i**) Bm vs. Ds, (**j**) Bm vs. Da, (**k**) Bm vs. Tg, (**l**) Pm vs. Da, (**m**) Ts vs. Ds, and (**n**) Ds vs. Da samples. Na: *Naja atra*, Bm: *Bungarus multicinctus*, Pm: *Protobothrops mucrosquamatus*, Ts: *Trimeresurus stejnegeri*, Ds: *Daboia siamensis*, Da: *Deinagkistrodon acutus*, and Tg: *Trimeresurus gracilis*.

**Table 1 biology-12-01206-t001:** Data summary of sequence reads for the individual oral samples of seven Taiwanese snake species. Na: *Naja atra*, Bm: *Bungarus multicinctus*, Pm: *Protobothrops mucrosquamatus*, Ts: *Trimeresurus stejnegeri*, Ds: *Daboia siamensis*, Da: *Deinagkistrodon acutus*, and Tg: *Trimeresurus gracilis*.

Sample Name	Raw HiFi Reads	Remove Primers	Filtered Reads	Denoised Reads	Non-Chimeric Reads
Na 1	20,241	19,179	18,345	17,940	17,910
Na 2	12,779	11,293	10,983	10,850	10,850
Na 3	13,112	11,003	10,559	10,285	9955
Na 4	11,740	11,070	10,800	10,709	10,709
Na 5	11,535	10,936	10,707	10,607	10,607
Na 6	13,057	12,333	11,975	11,817	11,817
Bm 1	13,652	12,611	12,226	11,951	11,940
Bm 2	10,806	10,388	10,135	10,029	10,029
Bm 3	12,306	11,213	10,882	10,607	10,607
Bm 4	15,043	13,337	13,048	12,915	12,915
Bm 5	10,436	9241	8990	8837	8837
Pm 1	15,813	14,108	13,732	13,690	13,690
Pm 2	14,113	12,000	11,405	11,252	11,039
Pm 3	12,350	10,178	9804	9763	9763
Pm 4	13,039	12,315	11,893	11,643	11,643
Pm 5	13,771	11,941	11,611	11,488	11,488
Pm 6	13,372	11,610	11,327	11,197	11,197
Ts 1	24,256	23,659	22,684	22,589	22,589
Ts 2	13,490	12,723	12,349	12,269	12,269
Ts 3	15,999	15,147	14,739	14,605	14,605
Ts 4	12,484	11,839	11,568	11,507	11,507
Ts 5	21,205	19,912	19,379	19,225	19,225
Ds 1	13,793	12,318	11,983	11,854	11,852
Ds 2	15,176	13,649	13,229	13,175	13,175
Ds 3	13,422	13,166	12,739	12,433	12,298
Ds 4	14,308	13,084	12,734	12,558	12,546
Ds 5	12,141	11,740	11,429	11,334	11,334
Da 1	13,651	12,426	11,964	11,819	11,816
Da 2	13,395	13,032	12,657	12,619	12,606
Da 3	13,200	12,489	12,058	11,655	11,655
Da 4	11,583	10,217	9996	9910	9910
Da 5	11,517	10,151	9949	9890	9890
Tg 1	13,637	12,419	12,044	11,886	11,886
Tg 2	12,071	10,967	10,690	10,535	10,535
Tg 3	12,850	11,453	11,149	10,827	10,827
Tg 4	13,424	11,993	11,712	11,549	11,549
Tg 5	13,913	12,609	12,320	12,039	12,039

**Table 2 biology-12-01206-t002:** Numbers of discovered non-chimeric reads and taxa for the oral samples from seven Taiwanese snake species. Na: *Naja atra*, Bm: *Bungarus multicinctus*, Pm: *Protobothrops mucrosquamatus*, Ts: *Trimeresurus stejnegeri*, Ds: *Daboia siamensis*, Da: *Deinagkistrodon acutus*, and Tg: *Trimeresurus gracilis*.

	Na	Bm	Pm	Ts	Ds	Da	Tg
Reads	71,848	54,328	68,820	80,195	61,205	55,877	56,836
Phylum	8	8	6	7	8	9	13
Class	22	20	21	21	23	24	30
Order	39	37	37	42	45	45	55
Family	69	70	72	69	95	81	103
Genus	115	115	124	121	226	155	246
Species	163	207	204	192	478	253	471

**Table 3 biology-12-01206-t003:** Characteristics (highlighted in blue: aerobic Gram-positive bacteria; red: aerobic Gram-negative bacteria; green: anaerobic bacteria) and relative abundances (in parentheses) of the top ten bacterial taxa present in oral cavities of each snake species. Na: *Naja atra*, Bm: *Bungarus multicinctus*, Pm: *Protobothrops mucrosquamatus*, Ts: *Trimeresurus stejnegeri*, Ds: *Daboia siamensis*, Da: *Deinagkistrodon acutus*, and Tg: *Trimeresurus gracilis*.

Order	Na	Bm	Pm	Ts	Ds	Da	Tg
**1**	* Bordetella * * trematum * (16.11%) ^1^	* Haemophilus * * felis * (29.60%)	* Clostridium * * innocuum * (31.73%)	* Puia * * dinghuensis * (36.04%)	* Clostridium * * dakarense * (10.71%)	* Morganella * * morganii * (29.12%)	* Cupriavidus * * numazuensis * (22.37%)
**2**	* Mycoplasma * * fastidiosum * (12.43%)	* Stenotrophomonas * * maltophilia * (15.23%)	* Cupriavidus * * numazuensis * (11.77%)	* Bacteroides * * fragilis * (21.40%)	* Cupriavidus * * numazuensis * (9.47%)	* Providencia * * rustigianii * (14.96%)	* Mycoplasma * * fastidiosum * (14.90%)
**3**	* Phocoenobacter * * uteri * (12.21%)	* Puia * * dinghuensis * (10.74%)	* Puia * * dinghuensis * (11.58%)	* Citrobacter * * freundii * (17.14%)	* Salmonella * * enterica enterica * serovar Typhimurium (4.63%)	* Puia * * dinghuensis * (12.63%)	* Pseudomonas * * aeruginosa * (9.39%)
**4**	* Pseudomonas * * helleri * (11.83%)	* Mycoplasma * * fastidiosum * (7.93%)	* Klebsiella * * pneumoniae * (9.89%)	* Yokenella * * regensburgei * (6.13%)	* Sphingomonas * * paucimobilis * (4.34%)	* Mycoplasma * * fastidiosum * (6.68%)	* Puia * * dinghuensis * (6.69%)
**5**	* Bisgaardia * * hudsonensis * (11.21%)	* Mycoplasma * * iguana * (4.82%)	* Clostridium * * cylindrosporum * (8.14%)	* Stenotrophomonas * * maltophilia * (5.46%)	* Bacteroides * * fragilis * (4.20%)	* Rheinheimera * * pacifica * (5.33%)	* Sphingomonas * * paucimobilis * (4.50%)
**6**	* Puia * * dinghuensis * (6.58%)	* Bergeyella * * zoohelcum * (4.56%)	* Sphingobacterium * * detergens * (2.88%)	* Morganella * * morganii * (2.29%)	* Paracoccus * * chinensis * (3.90%)	* Cupriavidus * * numazuensis * (4.14%)	* Sediminibacterium * * lactis * (2.45%)
**7**	* Proteiniphilum * * acetatigenes * (4.63%)	* Achromobacter * * insuavis * (4.00%)	* Luteibacter * * anthropic * (2.39%)	* Phyllobacterium * * myrsinacearum * (1.86%)	* Stenotrophomonas * * pavanii * (3.86%)	* Clostridium * * innocuum * (2.09%)	* Latilactobacillus * * curvatus * (1.50%)
**8**	* Serratia * * marcescens * (2.33%)	* Acinetobacter * * bereziniae * (3.13%)	* Desulfovibrio * * desulfuricans * (2.18%)	* Beijerinckia * * fluminensis * (1.71%)	* Chryseobacterium * * salipaludis * (3.24%)	* Rheinheimera * * hassiensis * (1.90%)	* Comamonas * * testosterone * (1.45%)
**9**	* Morganella * * morganii * (1.74%)	* Clostridium * * cylindrosporum * (2.78%)	* Epilithonimonas * * lactis * (2.01%)	* Cupriavidus * * numazuensis * (0.80%)	* Mammaliicoccus * * sciuri * (2.70%)	* Acinetobacter * * proteolyticus * (1.85%)	* Paracoccus * * suum * (1.35%)
**10**	* Haoranjiania * * flava * (1.64%)	* Comamonas * * testosterone * (2.12%)	* Oceanisphaera * * ostreae * (0.94%)	* Tepidimonas * * aquatica * (0.77%)	* Acinetobacter * * variabilis * (2.35%)	* Rheinheimera * * aquimaris * (1.61%)	* Paenarthrobacter * * nitroguajacolicus * (1.35%)

^1^ The relative abundance was calculated from the assigned read counts for each bacteria species divided by the total assigned read counts for a single snake species.

**Table 4 biology-12-01206-t004:** The 24 pathogenic (to humans) bacterial species and 24 non-pathogenic bacterial species among the top 10 abundant bacterial species of seven Taiwanese snake species. G+: Gram-positive, G−: Gram-negative, An: anaerobic bacteria.

Pathogen Bacterial Species	Non-Pathogen Bacterial Species
*Achromobacter insuavis* (G−)	*Beijerinckia fluminensis* (G−)
*Acinetobacter bereziniae* (G−)	*Chryseobacterium salipaludis* (G−)
*Acinetobacter proteolyticus* (G−)	*Clostridium cylindrosporum* (G+, An)
*Acinetobacter variabilis* (G−)	*Clostridium dakarense* (G+, An)
*Bacteroides fragilis* (G−, An) ^1^	*Cupriavidus numazuensis* (G−) ^1^
*Bergeyella zoohelcum* (G−)	*Epilithonimonas lactis* (G−)
*Bisgaardia hudsonensis* (G−)	*Haemophilus felis* (G−) ^1^
*Citrobacter freundii* (G−) ^1^	*Haoranjiania flava* (G−)
*Clostridium innocuum* (G+, An) ^1^	*Latilactobacillus curvatus* (G+, An)
*Comamonas testosterone* (G−)	*Mycoplasma fastidiosum* (G−) ^1^
*Desulfovibrio desulfuricans* (G−, An)	*Mycoplasma iguana* (G−)
*Klebsiella pneumoniae* (G−)	*Oceanisphaera ostreae* (G−)
*Morganella morganii* (G−) ^1^	*Paracoccus chinensis* (G−)
*Pseudomonas aeruginosa* (G−)	*Phocoenobacter uter* (G−)
*Serratia marcescens* (G−)	*Proteiniphilum acetatigenes* (G−, An)
*Luteibacter anthropi* (G−)	*Pseudomonas helleri* (G−)
*Mammaliicoccus sciuri* (G+)	*Puia dinghuensis* (G−) ^1^
*Paenarthrobacter nitroguajacolicus* (G+)	*Rheinheimera aquimaris* (G−)
*Phyllobacterium myrsinacearum* (G−)	*Rheinheimera hassiensis* (G−)
*Providencia rustigianii* (G−)	*Rheinheimera pacifica* (G−)
*Salmonella enterica enterica* serovar Typhimurium (G−)	*Sediminibacterium lactis* (G−)
*Sphingomonas paucimobilis* (G−)	*Sphingobacterium detergens* (G−)
*Stenotrophomonas maltophilia* (G−) ^1^	*Stenotrophomonas pavanii* (G−)
*Yokenella regensburgei* (G−)	*Tepidimonas aquatica* (G−)

^1^ This species belongs to the top 10 abundant bacterial species in all snake species.

## Data Availability

Data are contained within the article or Appendix A.

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
