# Peer review of "Comparisons of the Oral Microbiota from Seven Species of Wild Venomous Snakes in Taiwan Using the High-Throughput Amplicon Sequencing of the Full-Length 16S rRNA Gene"

_biology, 2023, doi:10.3390/biology12091206_

Round 1

Reviewer 1 Report

The move to long read microbiome analysis is excellent.  Do you have any similar bite wound analysis in prospect that could flag particular bite components as significant?

Author Response

The move to long read microbiome analysis is excellent.  Do you have any similar bite wound analysis in prospect that could flag particular bite components as significant?

Ans: Thank you for your positive reviews. To the best of our knowledge, there is no similar (long read microbiome) bite wound analysis being reported in previous literatures. Long-read microbiome analyses of bite wound samples from patients need to be carried out in future studies to flag significant bite components. We have added it at lines 469−470.

Reviewer 2 Report

Dear Authors, 

I find this research to be quite interesting and well done. However, to further enhance its quality, I kindly suggested that the authors address the comments provided.

Regards,

Author Response

I find this research to be quite interesting and well done. However, to further enhance its quality, I kindly suggested that the authors address the comments provided.

Ans: Thank you for your positive reviews. We have addressed the comments from all reviewers.

Reviewer 3 Report

Morbidity and mortality from snakebites are also linked to secondary infection by pathogenic bacteria found in poisonous snakes' oral microbiome. There is, however, a lack of knowledge on the whole microbiota in the oral cavity of Taiwanese snake species. Then, the present study investigated the high-throughput sequencing (next-generation sequencing; NGS) of the full-length 16S rRNA gene to identify the full compositions of microbiota within the oral cavity of seven Taiwanese snake species [Naja atra, Bungarus multicinctus, Protobothrops mucrosquamatus, Trimeresurus stejnegeri, Daboia siamensis, Deinagkistrodon acutus, and alpine Trimeresurus gracilis] belonging to the Viperidae and Elapidae families. The authors demonstrated for the first time that oral microbiota differed significantly between Viperidae and Elapidae snakes, as well as between Naja atra and Bungarus multicinctus within Elapidae, while bacterial species overlapped significantly within Viperidae. In my perspective, the main idea presented here is attractive, and all of the experiments confirm the findings. 

The English language is appropriate, in my opinion.

Author Response

Morbidity and mortality from snakebites are also linked to secondary infection by pathogenic bacteria found in poisonous snakes' oral microbiome. There is, however, a lack of knowledge on the whole microbiota in the oral cavity of Taiwanese snake species. Then, the present study investigated the high-throughput sequencing (next-generation sequencing; NGS) of the full-length 16S rRNA gene to identify the full compositions of microbiota within the oral cavity of seven Taiwanese snake species [Naja atra, Bungarus multicinctus, Protobothrops mucrosquamatus, Trimeresurus stejnegeri, Daboia siamensis, Deinagkistrodon acutus, and alpine Trimeresurus gracilis] belonging to the Viperidae and Elapidae families. The authors demonstrated for the first time that oral microbiota differed significantly between Viperidae and Elapidae snakes, as well as between Naja atra and Bungarus multicinctus within Elapidae, while bacterial species overlapped significantly within ViperidaeIn my perspective, the main idea presented here is attractive, and all of the experiments confirm the findings.
Comments on the Quality of English Language: The English language is appropriate, in my opinion.

Ans: Thank you for your positive reviews.

Reviewer 4 Report

In this study, authors reported the profiling of oral microbiota from seven venomous snake species in Taiwan, using the high-throughput amplicon sequencing technique on the full-length 16S rRNA gene of bacteria (collected from oral cavity of snakes sampled in as soon as possible after capture from the wild). This is by far the first study using this technique to study the oral microbiodata for all seven wild venomous snakes of Taiwan origin, hence it has the merit and novelty worthy for publication. The study is well designed and executed, the results are well analyzed and interpreted. There are limitations within the study which have also been acknowledged by the authors. On the whole I read it with much pleasure and happy to know that such study is carried out by these experts for the good of veterinary medicine and clinical therapy of venomous snakebites which complicate tissue infection, as seen in many cases of snakebites in Asia, especially those for cobras. On the whole there is no much issues except a few minor suggestions for authors consideration to polish up the writeup before its acceptance. 

1. Abstract: Line 28: I think there's no need of the round brackets ( ) enclosing the word Taiwanese. 

2. In this version (PDF) which I am reading, edits (track-change form) still remain. Please edit accordingly for a clean version later. 

3. Please standardize full-length throughout (instead of full length).

4. In all tables, suggest to include abbreviations for the genus and species (such as, Na: Naja atra; Bm: Bungarus multicinctus.... and so on)

5. Regarding "relative abundances" (%) in Table 3 and other bar chart figures as well as in the results and discussion - Can authors include in the method a brief description of the determination or estimation of the "abundance"? Was it deduced from the level of the sequences, or the number of the genes identified? It would be good to also indicate this as a footnote under the table. 

6. Table 4:  Salmonella enterica enterica Serovar Typhimurium -- please check if the "serovar" should be just small letter. 

7. The discussion is comprehensive with deep insights into the diversity  of oral microbiodata of these snakes. Authors attempted to also correlate the finding with clinical (human) envenoming, and explained that there are also other confounding factors to be considered for any variation observed. In their study and the PCoA (Fig. 7), it is interesting to note that the Na and Bm data occupied very different positions. It is known that cobra bite almost always cause very severe tissue necrosis while krait bite rarely causes significant local tissue problem. Could this be related to the finding in this study and perhaps explained by the difference in the strains (pathogenic vs. non-pathogenic) between the two snake species? Similarly, among the vipers and pit vipers, some species such as Russell's viper seem to have less complication of local tissue necrosis; not sure if this could be discussed in line with the oral microbiodata findings? 

----End----- 

The English language is generally good. Readers should be able to understand the paper without much problems.

Author Response

In this study, authors reported the profiling of oral microbiota from seven venomous snake species in Taiwan, using the high-throughput amplicon sequencing technique on the full-length 16S rRNA gene of bacteria (collected from oral cavity of snakes sampled in as soon as possible after capture from the wild). This is by far the first study using this technique to study the oral microbiodata for all seven wild venomous snakes of Taiwan origin, hence it has the merit and novelty worthy for publication. The study is well designed and executed, the results are well analyzed and interpreted. There are limitations within the study which have also been acknowledged by the authors. On the whole I read it with much pleasure and happy to know that such study is carried out by these experts for the good of veterinary medicine and clinical therapy of venomous snakebites which complicate tissue infection, as seen in many cases of snakebites in Asia, especially those for cobras. On the whole there is no much issues except a few minor suggestions for authors consideration to polish up the writeup before its acceptance.
Comments on the Quality of English Language: The English language is generally good. Readers should be able to understand the paper without much problems.

Ans: Thank you for your positive reviews.

  1. Abstract: Line 28: I think there's no need of the round brackets ( ) enclosing the word Taiwanese. 

Ans: We have removed the round brackets and revised the text in line 28.

  1. In this version (PDF) which I am reading, edits (track-change form) still remain. Please edit accordingly for a clean version later. 

Ans: We have removed all edits (track-change form) in the revised manuscript files.

  1. Please standardize full-length throughout (instead of full length).

Ans: We have standardized full-length throughout the manuscript.

  1. In all tables, suggest to include abbreviations for the genus and species (such as, Na: Naja atra; Bm: Bungarus multicinctus.... and so on)

Ans: We have added abbreviations for the scientific names of snakes in most tables and figures.

  1. Regarding "relative abundances" (%) in Table 3 and other bar chart figures as well as in the results and discussion - Can authors include in the method a brief description of the determination or estimation of the "abundance"? Was it deduced from the level of the sequences, or the number of the genes identified? It would be good to also indicate this as a footnote under the table. 

Ans: The estimation of the abundance was deduced from the assigned read counts of the sequences (= non-chimeric read counts − unassigned read counts); the relative abundance was calculated from the assigned read counts for each snake or bacteria species divided by the total assigned read counts. We have added it in the method (lines 167−171) and indicated it as a footnote in Table 3.

  1. Table 4:  Salmonella enterica enterica Serovar Typhimurium -- please check if the "serovar" should be just small letter. 

Ans: We have revised “Serovar” as “serovar”.

  1. The discussion is comprehensive with deep insights into the diversity  of oral microbiodata of these snakes. Authors attempted to also correlate the finding with clinical (human) envenoming, and explained that there are also other confounding factors to be considered for any variation observed. In their study and the PCoA (Fig. 7), it is interesting to note that the Na and Bm data occupied very different positions. It is known that cobra bite almost always cause very severe tissue necrosis while krait bite rarely causes significant local tissue problem. Could this be related to the finding in this study and perhaps explained by the difference in the strains (pathogenic vs. non-pathogenic) between the two snake species? Similarly, among the vipers and pit vipers, some species such as Russell's viper seem to have less complication of local tissue necrosis; not sure if this could be discussed in line with the oral microbiodata findings? 

Ans: Thank you for the comments. As you said, it is known that Na bite often causes severe tissue necrosis while Bm bite rarely causes significant local tissue problem. Compared to Ts, victims of Pm have higher rate of local necrosis or cellulitis. Victims of Ds, Da and Tg are rare in Taiwan and Ds bite seems to have less complication of local tissue necrosis. Tissue necrosis may be initiated by the action of snake toxins. How it is related to the differences of oral bacterial strains between Na and Bm or among Ds, Pm and other pitvipers remains further studies to verify the ability of the microbiota identified in this study to cause infection and necrosis. In addition, long-read microbiome analyses of bite wound samples from patients also need to be carried out in future studies. We have added related discussion (with references) at lines 427−434 and 469−470.